# Filamentation modulates allosteric regulation of PRPS

**Huan-Huan Hu[1†], Guang-Ming Lu[1†], Chia-Chun Chang[1†], Yilan Li[1], Jiale Zhong[1], Chen-Jun Guo[1], Xian Zhou[1], Boqi Yin[1], Tianyi Zhang[1], Ji-Long Liu[1,2]***

[1]School of Life Science and Technology, ShanghaiTech University, Shanghai, China; [2]Department of Physiology, Anatomy and Genetics, University of Oxford, Oxford, United Kingdom

**Abstract** Phosphoribosyl pyrophosphate (PRPP) is a key intermediate in the biosynthesis of purine and pyrimidine nucleotides, histidine, tryptophan, and cofactors NAD and NADP. Abnormal regulation of PRPP synthase (PRPS) is associated with human disorders, including Arts syndrome, retinal dystrophy, and gouty arthritis. Recent studies have demonstrated that PRPS can form filamentous cytoophidia in eukaryotes. Here, we show that PRPS forms cytoophidia in prokaryotes both in vitro and in vivo. Moreover, we solve two distinct filament structures of *E. coli* PRPS at near-atomic resolution using Cryo-EM. The formation of the two types of filaments is controlled by the binding of different ligands. One filament type is resistant to allosteric inhibition. The structural comparison reveals conformational changes of a regulatory flexible loop, which may regulate the binding of the allosteric inhibitor and the substrate ATP. A noncanonical allosteric AMP/ADP binding site is identified to stabilize the conformation of the regulatory flexible loop. Our findings not only explore a new mechanism of PRPS regulation with structural basis, but also propose an additional layer of cell metabolism through PRPS filamentation.

## Editor's evaluation

This paper provides new insights into how polymerization into two different structures modulates the activity of the enzyme PRPS. The molecular mechanisms proposed are supported by the data, and likely to be of general interest.

***For correspondence:**
liujl3@shanghaitech.edu.cn;
jilong.liu@dpag.ox.ac.uk

[†]These authors contributed equally to this work

**Competing interest:** The authors declare that no competing interests exist.

## Introduction

Phosphoribosyl pyrophosphate (PRPP) is an important intermediate in multiple metabolic pathways in cells. It is utilized in the biosynthesis of purine and pyrimidine nucleotides, histidine and tryptophan, and the cofactors NAD and NADP (*Hove-Jensen, 1988*; *Munch-Petersen, 1983*). PRPP is synthesized by PRPP synthase (PRPS), which catalyzes the transfer of diphosphate from ATP to ribose-5-phosphate (R5P), thereby generating AMP and PRPP (*Khorana et al., 1958*).

PRPS is essential for maintaining PRPP pool and cellular metabolic homeostasis. In humans, missense mutations of PRPP synthase isozyme 1 (PRPS1) alter enzyme activity or the allosteric regulation and are associated with many severe pathological outcomes (*Becker et al., 1995*; *Chen et al., 2013*; *Duley et al., 2011*). The decreased activity of this enzyme may result in neurological disorders, such as Arts syndrome, non-syndromic sensorineural deafness 2 (DFN2), Charcot-Marie-Tooth disease 5 (CMTX5), and retinal dystrophy. Conversely, hyperactivity of PRPS1 may lead to neurosensory defects, hyperuricemia, or gouty arthritis. Dysregulation of PRPS1 or PRPS2 activity and expression has been observed in many cancers and is associated with thiopurine resistance in recurrent childhood acute

lymphoblastic leukemia (*Cunningham et al., 2014*; *Li et al., 2015*). Therefore, the precise regulation of PRPS is of great significance in metabolism and physiology.

In general, organisms contain at least one PRPS gene. According to the biochemical properties, PRPS of different species can be divided into three categories. Class I PRPP synthase is the most widely distributed in phylogeny, including *E. coli* and humans. Phosphate ion (Pi) is necessary to activate class I PRPS, while ADP, an allosteric inhibitor, competes with Pi and ATP at allosteric and active sites, respectively (*Eriksen et al., 2000*; *Hove-Jensen et al., 1986*; *Nosal et al., 1993*; *Willemoës et al., 2000*). However, class II PRPS is active without Pi. ADP does not bind to the allosteric site of the class II PRPS but inhibits it competitively (*Krath et al., 1999*; *Krath and Hove-Jensen, 2001*). Class III PRPS present in archaea requires Pi to activate, but there is no allosteric mechanism (*Kadziola et al., 2005*).

It is reported that in prokaryotes and eukaryotes, there is a kind of micron level metabolic enzyme filament, which is called cytoophidium (cytoophidia for plural) (*Liu, 2016*; *Park and Horton, 2019*; *Zhou et al., 2020*). The cytoophidium is assembled by bundling filamentous polymers of metabolic enzymes (*Liu, 2016*). Dozens of cytoophidium-forming enzymes are identified in the genome-wide screening of budding yeast (*Noree et al., 2010*; *Shen et al., 2016*). In mammalian cells, the presence of CTP synthase (CTPS) and IMP dehydrogenase (IMPDH) cytoophidium is proposed to be correlated with the metabolic characteristics of specific tissues such as cancer and immune cells (*Calise et al., 2018*; *Chang et al., 2017*; *Duong-Ly et al., 2018*). The polymerization of human CTPS1 and IMPDH2 have been demonstrated to desensitize proteins to end-product inhibition or allosteric inhibition, indicating their physiological functions in tuning intracellular nucleotide levels (*Anthony et al., 2017*; *Lynch et al., 2017*). In addition, we have found asparagine synthase and proline synthesis enzyme P5CS form cytoophidia (*Zhang et al., 2020a*; *Zhang et al., 2018*; *Zhang et al., 2021*; *Zhong et al., 2022*). Recently, PRPS has been identified as a novel cytoophidium-forming enzyme in various eukaryotes, including budding yeast, fruit flies, zebrafish, and mammals (*Begovich et al., 2020*; *Noree et al., 2019*). The evolutionary conservation of PRPS filamentation implies the physiological roles of this structure. Moreover, PRPS, as the upstream enzyme of CTPS and IMPDH, indicates that the regulation of enzyme function through filamentation is particularly important for de novo nucleotide biosynthesis. Although the high-resolution structures of human, *Bacillus subtilis* and *E. coli* PRPS hexamers have been solved from several crystal forms (*Chen et al., 2015*; *Eriksen et al., 2000*; *Li et al., 2007*; *Zhou et al., 2019a*), the filamentation of PRPS remains largely unknown. Therefore, we aim to reveal the structure, function, and potential mechanism of PRPS filament.

In the present study, we find that *E. coli* PRPS can form filaments in vitro and in vivo. Using cryo-electron microscopy (cryo-EM), we solve two types of PRPS filaments with near-atomic resolution of 2.3–2.9 Å. Structural and biochemical analyses indicate that the formation of type A filament attenuates the allosteric inhibition. In addition, the conformational changes of the regulatory flexible loop (RF loop) suggest that it plays a role in the regulation of allosteric inhibition and ATP binding. A noncanonical allosteric binding site for AMP and ADP binding is identified in the type A filament, which participates in the regulation of the RF loop. Altogether, our findings reveal a novel mechanism of structural regulation of *E. coli* PRPS, and provide new insights into PRPS-related human disorders and potential clinical and industrial applications.

## Results

### PRPS hexamers assemble into two types of filaments

There is only one PRPS gene in *E. coli* genome, which has 47.5% sequence identity to human PRPS1 and PRPS2 (*Tatibana et al., 1995*). Recently, PRPS cytoophidium has been observed in various organisms, suggesting PRPS can form filamentous polymers. To elucidate the functions and potential molecular mechanism of PRPS filamentation, the structures of PRPS polymers were analyzed.

We expressed and purified *E. coli* PRPS from *E. coli* K12 strain in Transetta (DE3) cells, and analyzed its structures under cryo-EM. In order to determine the conditions for inducing PRPS polymerization, we incubated PRPS protein with its known ligands and examine it under negative staining electron microscopy. We provide cryo-EM data and model refinement statistics in *Table 1*.

ADP and Pi are well-known regulators of PRPS. Although ADP inhibits PRPS through allosteric inhibition and competitive inhibition, Pi competes with ADP at the allosteric site, which is required for

**Table 1.** Statistics of Cryo-EM structures in this study.

| | ecPRPS type A filament (EMD-33305, PDB 7XMU) | ecPRPS type B filament (EMD-33309, PDB 7XN3) | ecPRPS type A^AMP/ADP filament (EMD-33306, PDB 7XMV) |
|---|---|---|---|
| **Data collection and processing** | | | |
| EM equipment | Titan Krios | Titan Krios | Titan Krios |
| Detector | K3 camera | K3 camera | K3 camera |
| Magnification | 22,500 x | 22,500 x | 22,500 x |
| Voltage (kV) | 300 | 300 | 300 |
| Electron exposure ((e–/Å$^2$)) | 60 | 60 | 60 |
| Defocus range(μm) | –1.0 to –2.5 | –1.0 to –2.5 | –1.0 to –2.5 |
| Pixel size(Å) | 0.53 | 0.53 | 0.53 |
| Symmetry imposed | D3 | D3 | D3 |
| Number of collected movies | 3,474 | 3,131 | 2,566 |
| Initial particle images (no.) | 887,654 | 1186879 | 1066797 |
| Final particle images (no.) | 70,541 | 168,218 | 53,045 |
| Map resolution (Å) | 2.3 | 2.9 | 2.6 |
| FSC threshold | 0.143 | 0.143 | 0.143 |
| Map resolution range (Å) | 2.3–3.4 | 2.8–4.7 | 2.5–4.6 |
| **Refinement** | | | |
| Initial model used (PDB code) | 4S2U | 4S2U | 4S2U |
| Map sharpening B-factor(Å$^2$) | –45 | –98 | –51 |
| **Model composition** | | | |
| Non-hydrogen atoms | 15,294 | 14,016 | 15,228 |
| Protein residues | 1,842 | 1,830 | 1,842 |
| Ligands | ADP, HSX, PO4, MG | PO4 | AMP, HSX, ADP, MG |
| Waters | 822 | 54 | 810 |
| Ions | 18 | 12 | 12 |
| **B factors(Å$^2$)** | | | |
| Protein | 49 | 65 | 57 |
| Ligand | 55 | 67 | 58 |
| Water | 50 | 57 | 57 |
| **R.m.s. deviations** | | | |
| Bond lengths (Å) | 0.005 | 0.005 | 0.008 |
| Bond angles (°) | 0.736 | 0.609 | 0.800 |
| **Validation** | | | |
| MolProbity score | 1.48 | 2.11 | 1.58 |
| Clashscore | 4.23 | 6.13 | 5.31 |
| Poor rotamers (%) | 1.57 | 2.37 | 1.18 |
| **Ramachandran plot** | | | |
| Favored (%) | 97.36 | 92.08 | 96.37 |
| Allowed (%) | 2.64 | 7.92 | 3.63 |

*Table 1 continued on next page*

*Table 1 continued*

|  | ecPRPS type A filament | ecPRPS type B filament | ecPRPS type A$^{AMP/ADP}$ filament |
|  | (EMD-33305, PDB 7XMU) | (EMD-33309, PDB 7XN3) | (EMD-33306, PDB 7XMV) |
| --- | --- | --- | --- |
| Disallowed (%) | 0 | 0 | 0 |

catalysis. In addition, Mg$^{2+}$ is known to promote ATP binding at the active site (*Gibson and Switzer, 1980*).

When PRPS was incubated without ligands, no filaments were found, indicating that *E. coli* PRPS was not easy to polymerize. PRPS filaments were also not found with only AMP (2 mM). However, when PRPS was incubated only with ATP or any adenine nucleotide and Mg$^{2+}$ (10 mM), many PRPS filaments could be observed. On the other hand, we also incubated PRPS at Pi concentrations of 10, 30, and 50 mM, and found that PRPS filamentated at 50 mM Pi (*Figure 1—figure supplement 1*).

We speculate that the conformation of PRPS polymer may change with the binding of different ligands. Therefore, we selected two conditions for structural analysis using cryo-electron microscopy (cryo-EM) and single-particle analysis. The first condition is the combination of ATP (2 mM) + Mg$^{2+}$ (10 mM) and the second condition is only Pi (50 mM). These two conditions can induce PRPS filamentation without triggering the reaction, and the binding modes of most PRPS ligands are expected to be determined in the models. As a result, two distinct filament structures, type A (ATP and Mg$^{2+}$) and type B (Pi) filaments were solved (*Figure 1*, *Figure 1—figure supplements 2 and 3*). In both models, PRPS hexamers are stacked in rows to form filamentous polymers. The twist and rise of type A filament are 27° (left-handed twist) and 63 Å, respectively, and the twist and rise of type B are 46° (left-handed twist) and 66 Å, respectively (*Figure 1C and D*).

Surprisingly, in type A filament, we found R5P at the active site, and the ATP binding site was occupied by ADP rather than ATP. In addition, another ADP is located at a noncanonical binding site (allosteric site 2), which is bound by AMP in the *Legionella pneumophila* PRPS structure (PDB ID: 6NFE), while Pi is located at the binding position of the β-phosphate of ADP at the canonical ADP allosteric site (allosteric site 1) (*Figure 1A*, *Figure 1—figure supplement 4*).

Since we did not add ADP to the mixture for sample preparation, ADP in the model might come from the spontaneous hydrolysis of ATP or be preserved during protein purification. The latter scenario can also explain the unexpected presence of R5P in the model. Meanwhile, in type B filament, R5P binding site and allosteric site 1 are bound by Pi, while ATP binding site and allosteric site 2 are empty (*Figure 1B*).

The structural comparison between the two models shows that the PRPS monomers in type A and type B filaments are highly similar except for the regulatory flexible loop region (RF loop, Y94 to T109) (*Figure 2A*). The main difference between the hexamers in the two filaments lies in the relative position of the monomers in the parallel dimer. The 5.2° rotation of hexamer leads to closed (type A) and open (type B) conformations (*Figure 2B–D*, *Figure 2—video 1*).

## Ligand binding modes in type A filaments

In the type A filament model, R5P and ADP at the active site are in association with two Mg$^{2+}$. One Mg$^{2+}$ (Mg site 1) coordinates the C1, C2, and C3 hydroxyl groups of R5P with D170 and two water molecules, and the other Mg$^{2+}$ (Mg site 2) coordinates oxygens of the α- and β-phosphates of ADP (active site) with H131 and three water molecules (*Figure 3A and B*, *Figure 1—figure supplement 4*). The ADP at the active site of chain B forms hydrogen bonds with D37 in chain C, and there is a π-π interaction between F35 in chain C and adenine base. R99 and H131 in chain B form salt bridges with the β and α-phosphate, respectively (*Figure 3B and C*). The α-phosphate of the ADP also interacts with R5P through hydrogen bonds with the C-1 hydroxyl group. R5P of chain B forms hydrogen bonds with D170, D220, D221, T225, and T228 in the same chain (*Figure 3B and C*).

ADP binds allosteric site 2 in chain A through hydrogen bonding with R102 in chain B, and S149, E133 in chain A, and a π-π interaction between F147 of chain A and adenine bases, and salt bridges between R102 in chain B, R178 in chain A, and phosphates (*Figure 3D*, *Figure 1—figure supplement 5*). Allosteric site 2 binds to AMP in *Legionella pneumophila* PRPS structure (PDB ID: 6NFE), which is

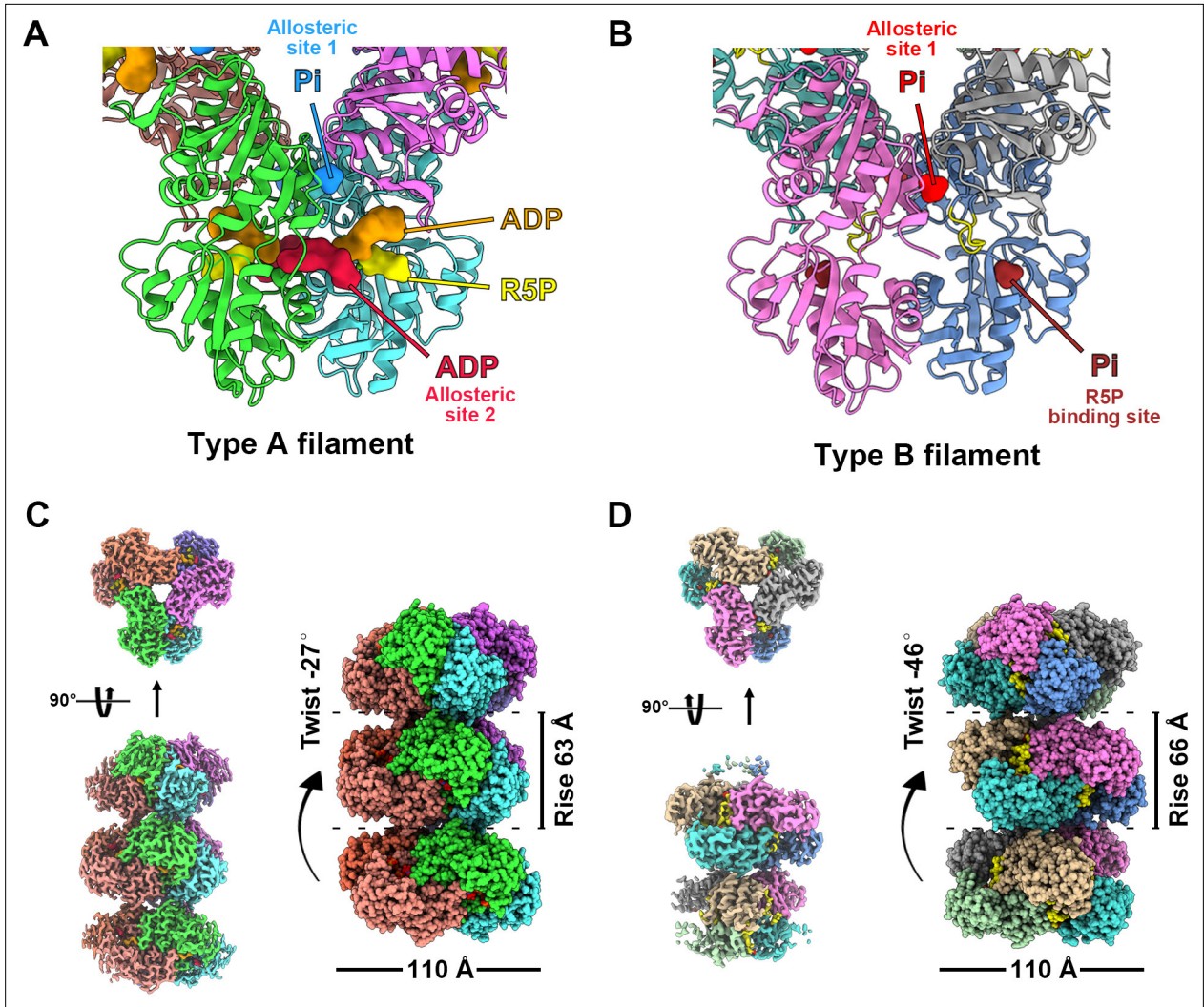

**Figure 1.** with 8 supplements. **Overall structures of** *E. coli* **PRPS type A and type B filaments**. (**A**) Type A filament ligands. The unit of *E. coli* PRPS type A filaments is hexamer with D3 symmetry. The hexamer has six identical ligand binding sites, one of which is shown in the figure. Different ligands are labeled with different colors. In type A filament, phosphate ion (Pi) binds to allosteric site 1, ADP binds to allosteric site 2 (red), and ATP binds to active site (brown). R5P also can be seen in the active site. (**B**) Type B filament ligands. The unit of type B filament is similar to that of type A filament, and one of the six identical ligand binding sites is shown here. In type B filament, the ATP binding site of active site is not bound by any ligand, while the R5P binding site and allosteric site 1 are bound by Pi. (**C**) Cryo-EM reconstruction of type A filament (C, 2.3 Å resolution). On the left is the electron density map of type A filament. On the right is the reconstruction structure of type A filament. The diameter and rise of type A filament are 110 Å and 63 Å, respectively. When hexamers are aggregated into type A filament, the adjacent hexamer us twisted by 27°. (**D**) Cryo-EM reconstructions of type B filament (2.9 Å resolution). On the left is electron density map of type A filament. On the right is the reconstruction structure of type A filament. The diameter of type B filament is same as that of type A filament. The rise and twist of type B filament are 66 Å and 46°, respectively.

The online version of this article includes the following figure supplement(s) for figure 1:

**Figure supplement 1.** *E. coli* PRPS is assembled into filaments in vitro.

**Figure supplement 2.** Cryo-EM data processing of PRPS type A filament.

**Figure supplement 3.** Cryo-EM data processing of PRPS type B filament.

**Figure supplement 4.** Electron density maps of type A and type A$^{AMP/ADP}$ in allosteric site and active site.

**Figure supplement 5.** Structure analysis of PRPS type A$^{AMP/ADP}$ filament.

**Figure supplement 6.** Cryo-EM data processing of PRPS type A$^{AMP/ADP}$ filament.

**Figure supplement 7.** Representative Cryo-EM density maps of individual regions of the PRPS type A filament.

**Figure supplement 8.** Representative Cryo-EM density maps of individual regions of the PRPS type B filament.

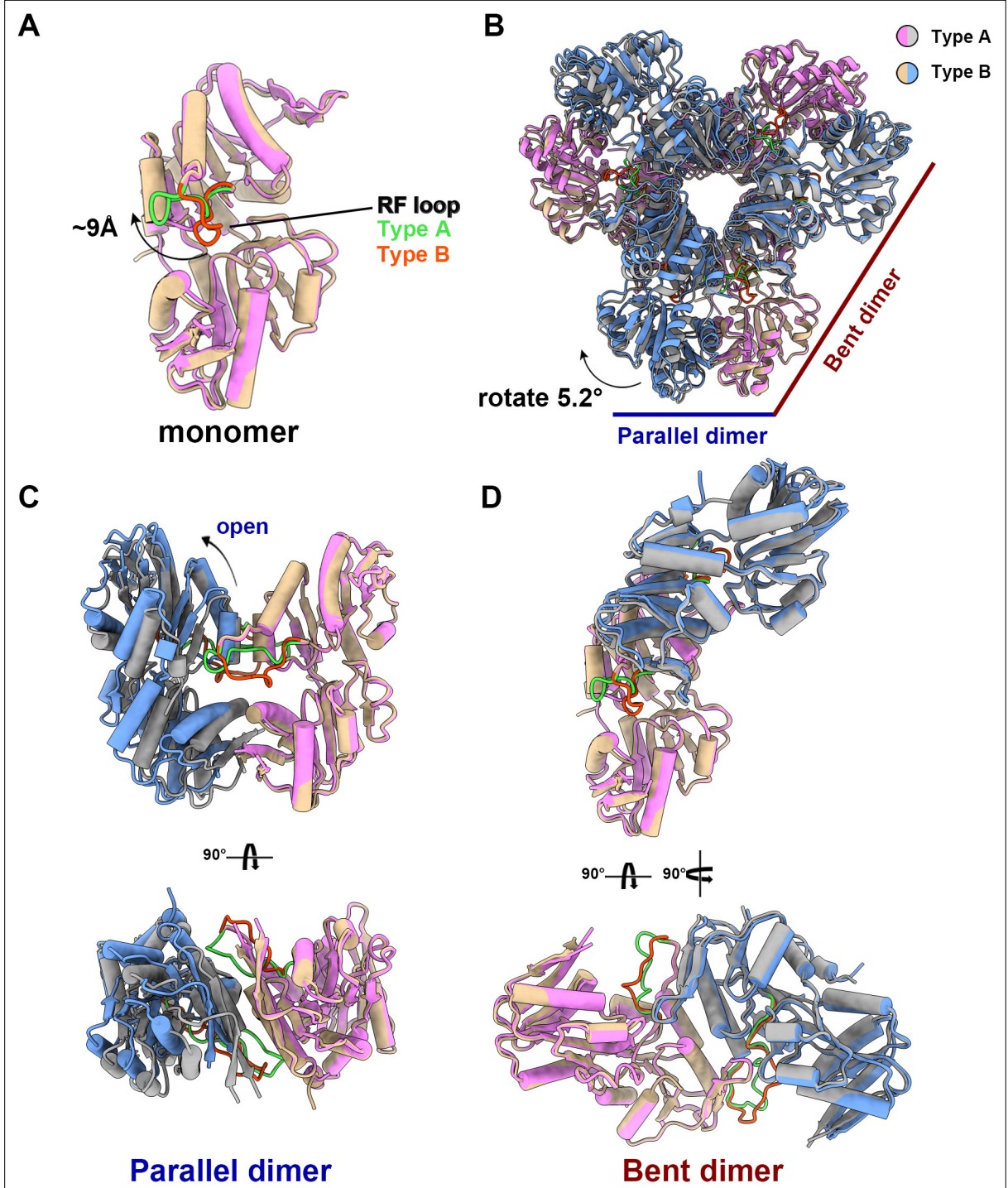

**Figure 2.** The structural comparison of *E. coli* PRPS type A and type B filaments. (**A**) Monomers in the closed (pink for type A polymer) and open (yellow for type B polymer) hexamers. In the monomer of type A and type B polymers, the RF loop is green and red respectively. The shift of RF loop is about 9 Å. (**B**) Structural comparison of hexamers of the type A (pink and gray) and type B (yellow and blue). Structural comparison of parallel and bent dimers in of type A (C, pink and gray) and type B (D, yellow and blue) filament hexamers.

The online version of this article includes the following video for figure 2:

**Figure 2—video 1.** Structural transition of ecPRPS hexamer conformations in type A and type B polymers.
https://elifesciences.org/articles/79552/figures#fig2video1

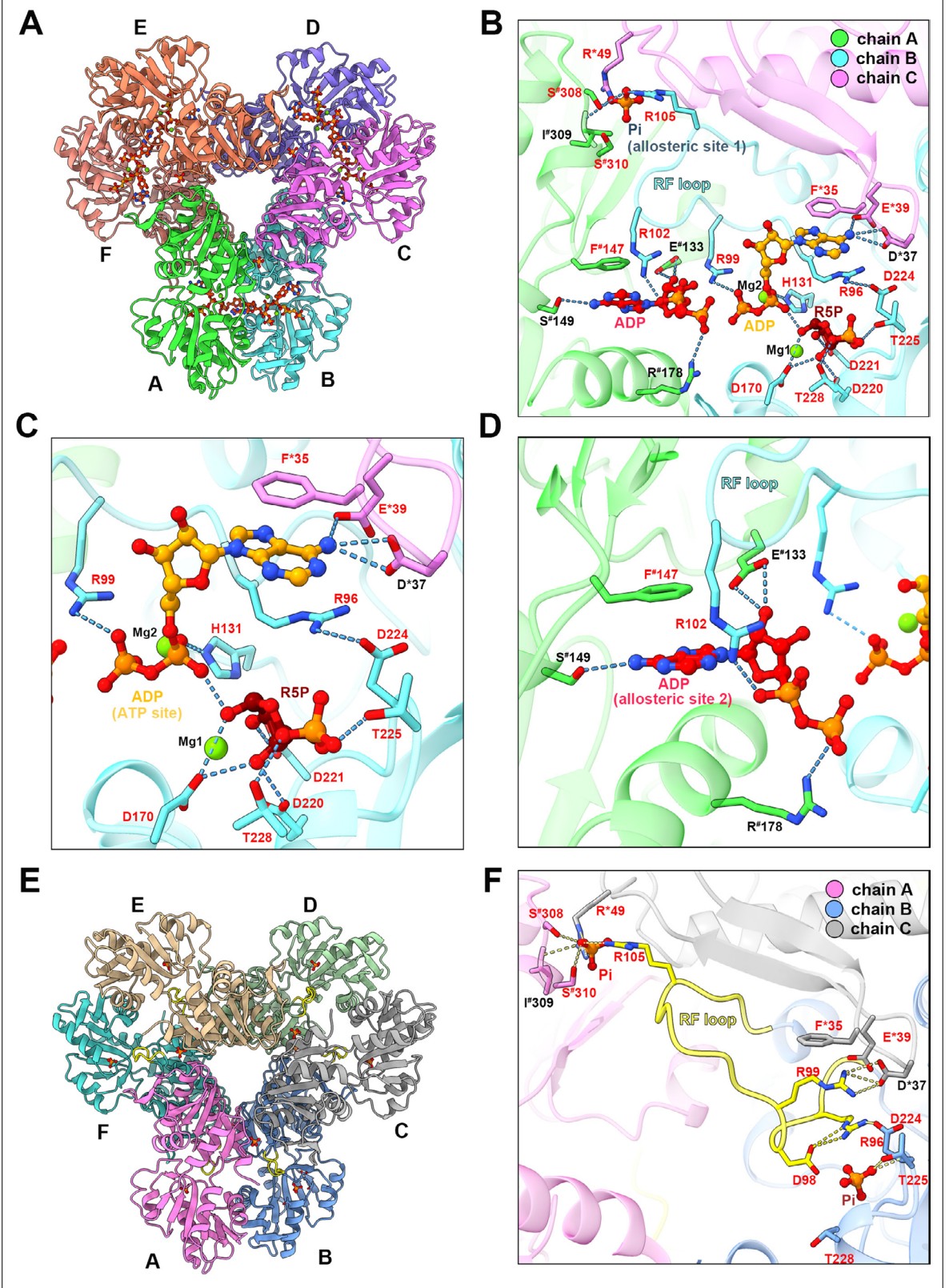

**Figure 3.** with 2 supplements. Ligand binding modes in *E. coli* PRPS type A and type B filaments. (**A**) Hexamer of type A filament. Each chain is marked with a different color. (**B**) ADP and R5P are identified on the active site of PRPS in type A filament, while allosteric site 1 is bound by Pi and allosteric site 2 is bound by ADP. The residues that interact with ligands are indicated. Residues in chain A number with the # symbol and in chain C number with the * symbol. Residues in red are conserved in various organisms. Each chain is marked with a different color(dash lines in cyan indicate hydrogen bonds).

*Figure 3 continued on next page*

*Figure 3 continued*

(**C**) Ligands of the active site of type A filament. ADP and Mg$^{2+}$ occupy ATP binding sites at active sites. R5P and Mg$^{2+}$ can also be seen in the active site. Residues in red are conserved in various organisms. Each chain is marked with a different color. (**D**) ADP in allosteric site 2 of type A filament. Residues in red are conserved in various organisms. Each chain is marked with a different color. (**E**) Hexamer of type B filament. Each chain is marked with a different color. (**F**) In type B filament, the ATP binding site of the active site is not bound by any ligand, while the R5P binding site and allosteric site 1 are bound by Pi. Residues in chain A number with the # symbol and in chain C number with the * symbol. Residues in red are conserved in various organisms. Each chain is marked with a different color(dash lines in yellow indicate hydrogen bonds).

The online version of this article includes the following figure supplement(s) for figure 3:

**Figure supplement 1.** Structure comparison of PRPS in various organisms.

**Figure supplement 2.** Comparison of PRPS sequences of various organisms.

adjacent to allosteric site 2 bound with SO4$^{2+}$ in multiple human PRPS1 structures (PDB ID: 2H06 and 2HCR) (*Figure 3—figure supplement 1*). Parallel to the above two structures, we also solved another type A filament (type A$^{AMP/ADP}$), which was formed under the conditions of AMP (2 mM)+ADP (2 mM) (*Figure 1—figure supplement 5*). This structure is almost the same as the former type A filament model, but its allosteric site 2 is bound by AMP rather than ADP (*Figure 1—figure supplement 5*). In addition, the allosteric site 1 in type A$^{AMP/ADP}$ model is empty, supporting that the presence of ADP and Pi in the former type A filament model were due to spontaneous hydrolysis of ATP. According to our two type A filament models, the allosteric site 2 of *E. coli* PRPS may accommodate both AMP and ADP. Allosteric site 2 bound with AMP/ADP in *E. coli* lies in space adjacent to allosteric site 2 bound with SO4$^{2+}$ in human PRPS1, suggesting that these two noncanonical allosteric sites are functionally related (*Figure 3—figure supplement 2*).

## Dynamic RF loop reveal novel regulatory mechanisms of PRPS

In the model of type B filament, allosteric site 1 is bound by Pi and allosteric site 2 is incomplete due to the conformational change of hexamer from closed to open (*Figure 3E and F*). The 5-phosphate of R5P on the active site is replaced by Pi (*Figure 3F*). Intriguingly, our density map shows that the ATP binding site in the active site is not bound by the ligand, but occupied by the RF loop (*Figure 4A*).

This conformation of the RF loop differs from previous structures including the *Bacillus substilis* PRPS (PDB ID: 1DKU) (*Figure 4B*, *Figure 3—figure supplement 1*). The B-factor of the RF loop in the type B filament ranges from 50.5 to 93.3 Å$^2$, indicating its flexibility (*Figure 4C*). However, in type A filaments, the RF loop is stabilized by a variety of interactions, including the salt bridge between R102 and α-phosphate of ADP at allosteric site 2, resulting in smaller B factors ranges from 29.4 to 44.5 Å$^2$ (*Figure 4C*).

On the other hand, the RF loop in type A filament is stabilized at a conformation and covers the ADP binding pocket at allosteric site 1. This conformation results in the closure of allosteric site 1, thereby preventing the binding of allosteric inhibitor ADP (*Figure 4D*). This may also explain the fact that ADP does not bind to allosteric site 1 in type A$^{AMP/ADP}$ filament, although its binding competitor Pi is absent (*Figure 1—figure supplement 5*).

## Distinct contacts of PRPS hexamers in type A and B filaments

In the two filament models, the relative positions of PRPS monomers of parallel dimers are different, resulting in different contact interfaces between the two PRPS hexamers. In type A filaments, hexamers are connected by salt bridges formed between R301 and E298 pairs, by the hydrogen bonds between R301, N305, and E307, and also by van der Waals force between R302 and R301 (*Figure 5A*). In the type B filament, the connection between hexamers relies on the π-cation interaction between Y24 and R22, and the hydrogen bond between R301 and L23 (*Figure 5B*). These different interfaces may prevent the polymerization of heterogenous hexamers.

In order to investigate the functions of these two types of filaments, we have generated PRPS$^{R302A}$ and PRPS$^{Y24A}$ respectively to disrupt the formation of type A and type B filaments. We also generated a mutant PRPS (PRPS$^{R302A/Y24A}$) carrying both R302A and Y24A mutations, which may not be able to form filaments. The ability of each PRPS mutant to form type A and type B filaments was evaluated under negative staining electron microscopy. As expected, PRPS$^{R302A/Y24A}$ failed to assemble filaments under all conditions, and PRPS$^{R302A}$ and PRPS$^{Y24A}$ could only form type B and type A filaments, respectively (*Figure 6A*).

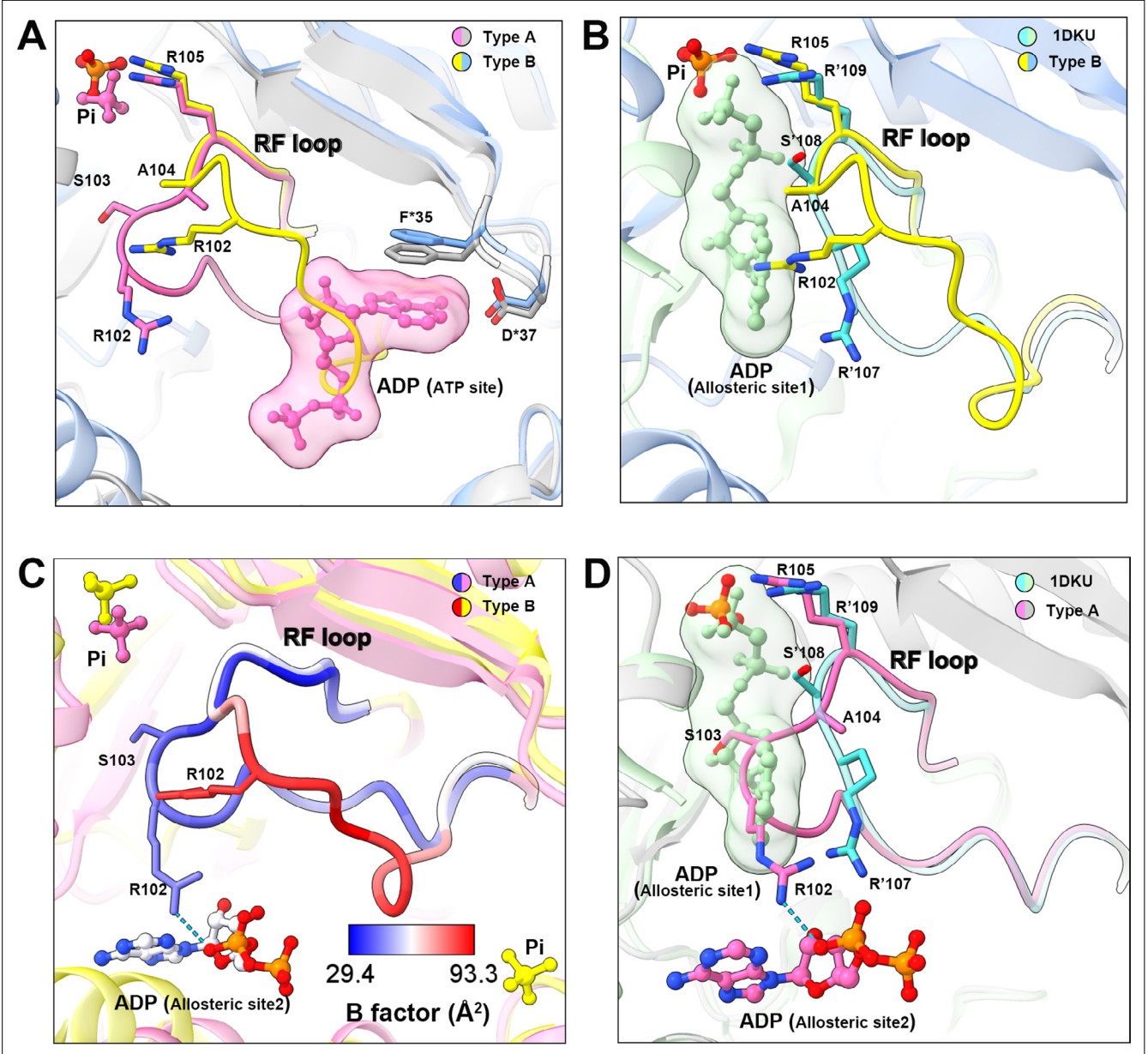

**Figure 4.** Conformational changes of the RF loop. (**A**) Comparison of RF loop structures in type A (pink and gray) and type B (yellow and blue) filaments. In type B filament, the RF loop partially occupies the active site that blocks nucleotide binding. (**B**) Comparison of RF loop structures between type B filament and *Bacillus substilis* PRPS (PDB ID: 1DKU). Residues in 1DKU number with the ' symbol. (**C**) B factors are shown on the RF loop of type A and type B filament models. (**D**) Structural comparison shows that the *Bacillus substilis* PRPS (PDB ID: 1DKU) model, the RF loop in type A filament overlaps with ADP at allosteric site 1. Residues in 1DKU number with the ' symbol.

## Filamentation regulates allosteric inhibition of PRPS

To elucidate the function of PRPS filamentation, the in vitro activity of PRPS was determined by coupling reaction. In the reaction mixture, the newly synthesized PRPP by PRPS would be subsequently utilized by phosphoribosyltransferase (OPRT) in the reaction *orotate (OA) +PRPP → orotidine 5'-monophosphate (OMP) +PPi* (*Krungkrai et al., 2005*). Therefore, the PRPP production could be measured by the consumption of OA, and the absorbance of OA is 295 nm.

In the absence of ADP, we found the substrate concentration used in the assay (0.6 mM ATP, 0.6 mM R5P), PRPS catalyzed the reaction at the maximum rate (*Figure 6—figure supplement 1A,B*). Pi is known as an activator of class I PRPS. PRPS production could not be detected without Pi (*Figure 6— figure supplement 1C*). When different concentrations of Pi were added to the reaction mixtures,

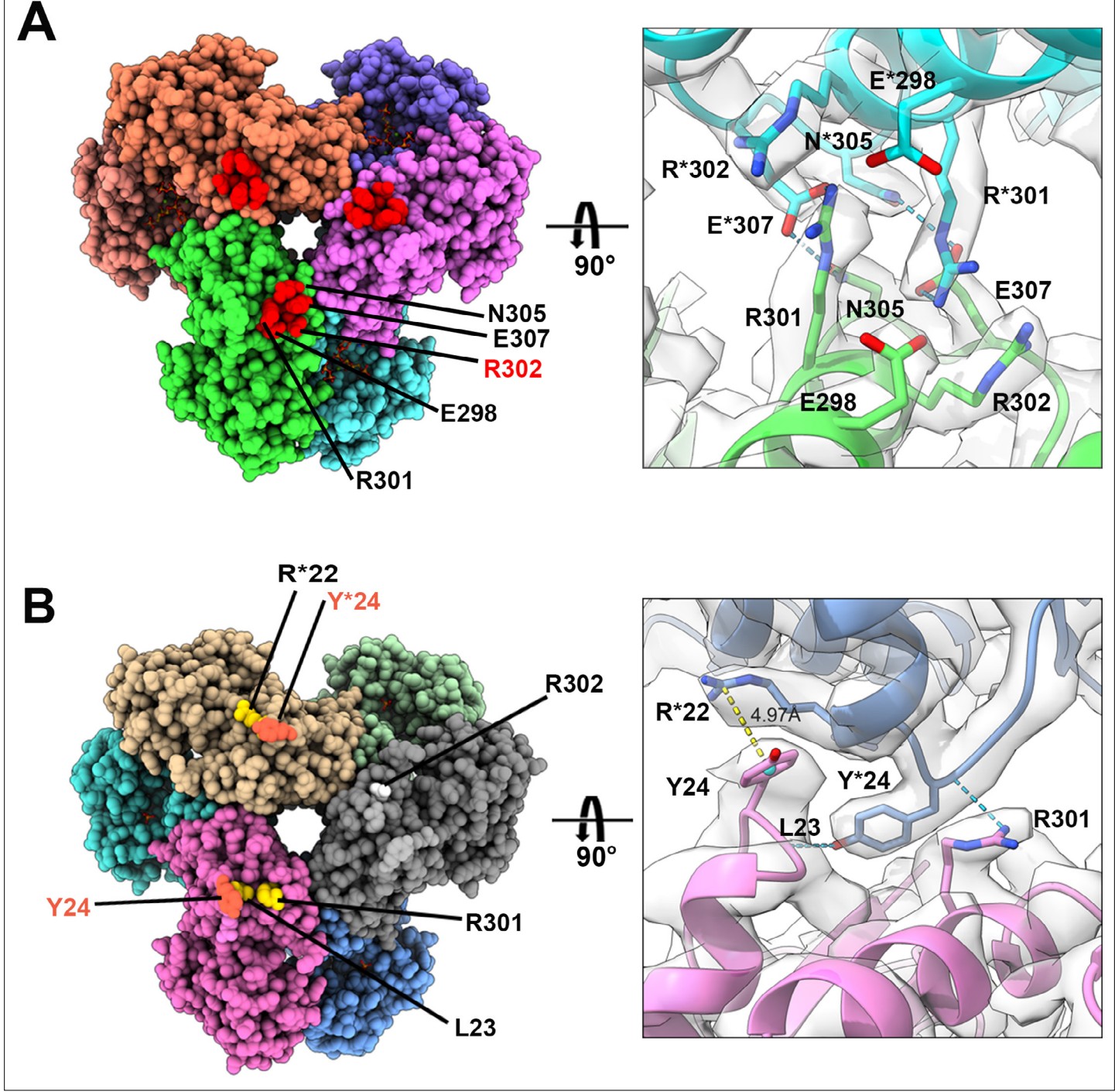

**Figure 5.** Distinct contacts of hexamers in *E. coli* PRPS type A and B filaments. (**A and B**) Maps and models of type A (**A**) and type B (**B**) filaments reveal distinct interfaces between adjacent hexamers in the two types of filaments. Residues responsible for the interactions are indicated. Residues in another hexamer number with the * symbol.

the activity peaked at 5–10 mM and gradually decreased at higher concentrations (*Figure 6—figure supplement 1C*). Therefore, we used 0.6 mM ATP, 0.6 mM R5P, and 10 mM Pi and different concentrations of ADP to analyze the activity of each PRPS mutants.

In the absence of ADP, the activity of PRPS[R302A] decreased significantly, while the activity of PRPS[R302A/Y24R] was higher (*Figure 6B*). When ADP (0.1 mM) was added into the mixture, the activity of PRPS[R302A] and PRPS[R302A/Y24A] dropped dramatically by 82.3% and 85.1% respectively, whereas the

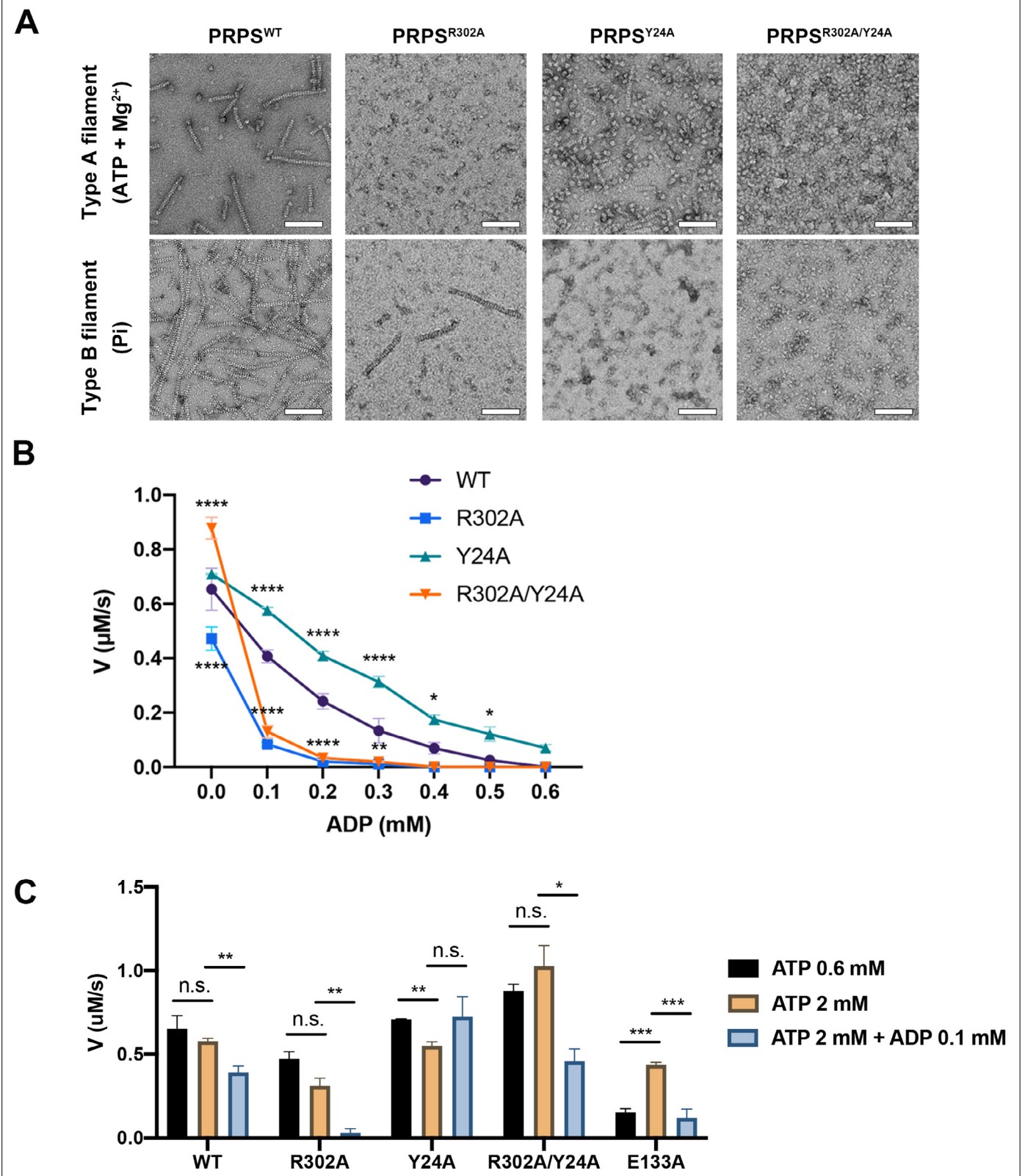

**Figure 6.** Filamentation regulates allosteric inhibition of *E. coli* PRPS. (**A**) Under the condition of inducing type A and type B filaments, the negative staining electron microscopic images of wild-type and mutant *E. coli* PRPS. Scale bars = 100 nm. (**B**) The graph shows the catalytic activity of wild-type and mutant *E. coli* PRPS with various amounts of ADP (Tukey's test). (**C**) Bar graph shows the catalytic activity of wild-type and mutant *E. coli* PRPS in the

*Figure 6 continued on next page*

*Figure 6 continued*

reaction mixtures containing different amounts of ATP and ADP (Student's *t-test*). Error bars = standard error of the mean (S.E.M.). *p<0.05, ** p<0.01, *** p<0.001 and **** p<0.0001.

The online version of this article includes the following figure supplement(s) for figure 6:

**Figure supplement 1.** Catalytic activity of *E. coli* PRPS with different concentrations of ligands.

activities of PRPS$^{WT}$ and PRPS$^{Y24A}$ decreased only by 37.8% and 18.9%, respectively (*Figure 6B*). With the increase of ADP concentration, the activity of each group decreased gradually. However, PRPS$^{R302A}$ and PRPS$^{R302A/Y24R}$ were nearly inactive when ADP concentration was higher than 0.2 mM.

ADP inhibits PRPS through allosteric and competitive inhibition. The inhibition we observed in these conditions may be a combination of these two mechanisms. To reduce competitive inhibition, we then increased the ATP concentration to 2 mM (*Figure 6C*). When ADP was not present, the increase of ATP concentration did not change the activity of all PRPS. However, when ADP (0.1 mM) was supplied, activity deceased in most groups (*Figure 6C*). The activity of PRPS$^{WT}$ and PRPS$^{R302A}$ dropped by 32.2% and 91.6%, respectively. The only exception is PRPS$^{Y24A}$, whose activity has not changed, indicating that the absence of type B filament leads to significant resistance to allosteric inhibition. After adding ADP, the activity of PRPS$^{R302A/Y24R}$ decreased by 55.4%, indicating that filamentation is not required for allosteric inhibition. Although it is unlikely that all PRPS were within filaments during the measurement, these results still show that polymerization significantly affects the activity of PRPS under these in vitro conditions. Our findings on conformational changes of RF loop may explain the resistance of type A filaments to allosteric inhibition of ADP.

## AMP/ADP at the allosteric site 2 facilitate ATP binding

The inhibitory function of the allosteric site 1 is well-known, although the molecular mechanism is still uncertain (*Hove-Jensen et al., 2017*). The function of the allosteric site 2, however, is largely unclear. According to our model, allosteric site 2 can be bound by AMP and ADP. It is worth noting that AMP is also one of the products in the reaction. The type A$^{AMP/ADP}$ filament model suggests that allosteric site 2 may favor AMP over ADP. That is, allosteric site 2 may bind to AMP under all reacting conditions. Nevertheless, we measured PRPS activity at different AMP levels and found no significant difference among groups (*Figure 6—figure supplement 1D*).

Our model suggests that AMP/ADP at allosteric site 2 may contribute in the stabilization of the RF loop through the interaction between R102 and α-phosphate of ADP or AMP. Since the stabilization of the RF loop is expected to prevent ADP from entering allosteric site 1 and avoid the interference RF loop on ATP binding, we suspect that AMP/ADP at allosteric site 2 may enhance PRPS activity. Therefore, we intended to impair the interaction between the C2 hydroxyl group of ADP and the side chain of E133 by introducing the point mutation E133A. As expected, when ATP was 0.6 mM, the activity of PRPS$^{E133A}$ was four times lower than that of PRPS$^{WT}$. However, when ATP was increased to 2 mM, the activity of PRPS$^{E133A}$ increased to a level comparable to that of PRPS$^{WT}$. Corresponding human PRPS1$^{S132A}$ mutant also displayed reduced activity in a previous study (*Li et al., 2007*). These results suggest that AMP/ADP at allosteric site 2 may facilitate the binding of ATP at the active site (*Figure 6C*).

## *E. coli* PRPS filaments assemble into cytoophidia in vivo

Filamentous polymers of various metabolic enzymes have been shown to bundle into a large filamentous structure, the cytoophidium, in a broad spectrum of organisms. PRPS cytoophidia have been observed in multiple eukaryotes including mammals. We fused the sequences of PRPS$^{WT}$ and PRPS$^{R302A}$, PRPS$^{Y24A}$ and PRPS$^{R302A/Y24A}$ mutants with mCherry and overexpressed them in *E. coli*, respectively. In a small subset of cells (less than 1%), filamentous structures of PRPS$^{WT}$-mCherry could be observed (*Figure 7A*). In contrast, PRPS$^{R302A}$-mCherry filaments and punctate aggregates were observed in about 1% cells. In cells expressing PRPS$^{Y24A}$-mCherry, filaments were not found, but punctate aggregates were observed less than 1% cells. At all stages of growth, no filaments or punctate aggregates were observed in cells expressing PRPS$^{R302A/Y24A}$-mCherry.

It is worth noting that the size of PRPS polymer may be within tens of nanometers, and the filaments or dots we observed with fluorescence microscope may be in a much larger scale. All forms of

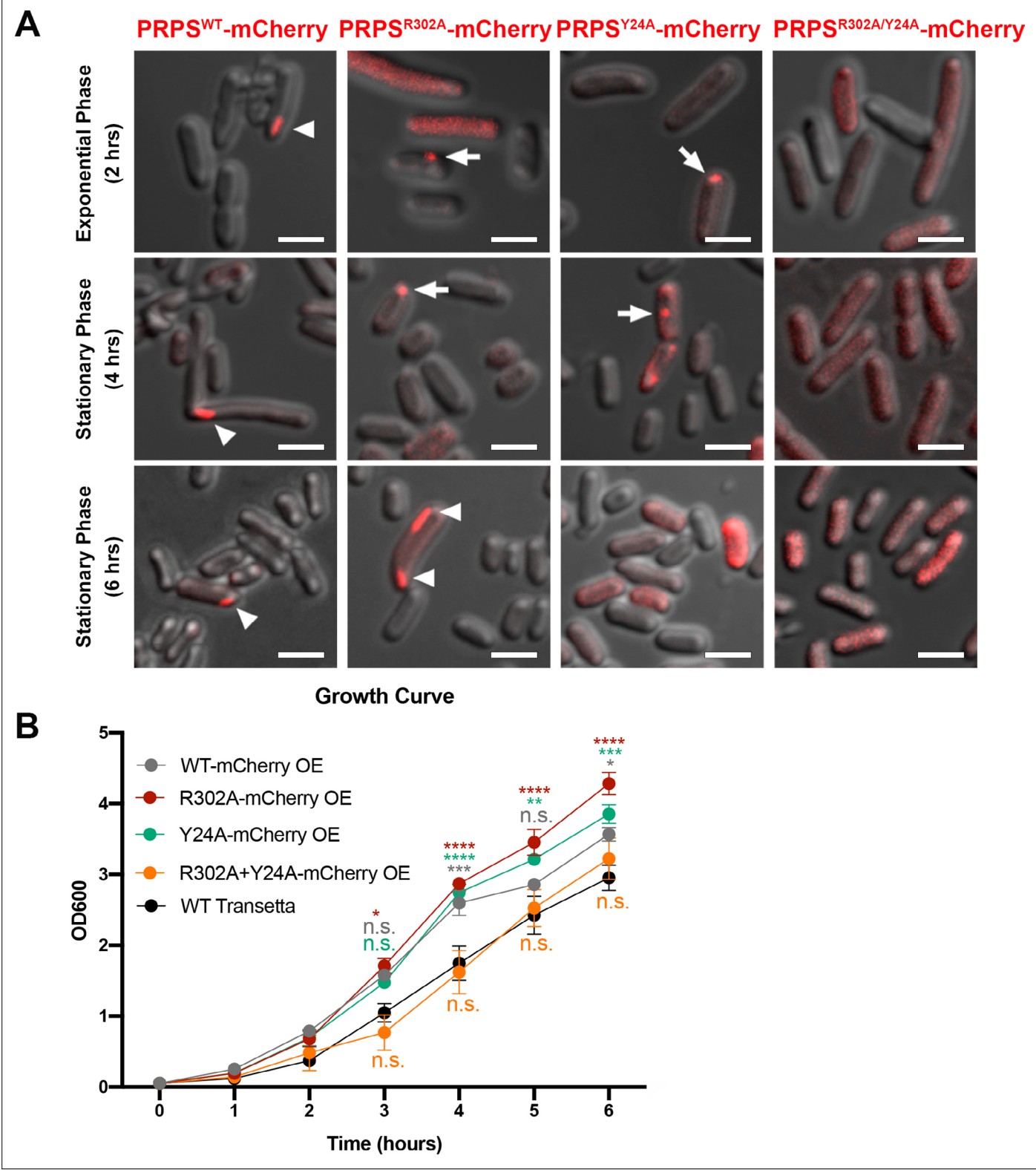

**Figure 7.** *E. coli* PRPS forms type A and type B filaments in vivo. (**A**) Representative images of Transetta *E. coli* strains overexpressing wild-type PRPS-mCherry and mutant PRPS-mCherry fusion proteins. Filamentous cytoophidia and punctate aggregates are indicated by arrowheads and arrows, respectively. Scale bars = 2 µm. (**B**) Growth curves of wild-type Transetta cells and cells overexpressing wild-type PRPS-mCherry and mutant PRPS-mCherry fusion proteins (Tukey's test). Error bars = S.E.M. *p<0.05, ** p<0.01, *** p<0.001 and **** p<0.0001.

aggregates, filaments, and dots contain PRPS polymers, and dispersed polymers may also be present in cells without detectable aggregates.

Next, we analyzed the growth curve of *E. coli* overexpressing PRPS-mCherry proteins. The growth rate of cells overexpressing PRPS[WT], PRPS[R302A], and PRPS[Y24A] was significantly faster than that of wild-type Transetta cells, while the growth rate of cells overexpressing PRPS[R302A/Y24A] was similar to that of wild-type cells (*Figure 7B*). Although we cannot rule out the unexpected effects of PRPS overexpression and mCherry tagging, our results show that both type A and type B PRPS filaments exist in vivo.

## Discussion

The assembly of filamentous polymers has emerged as a common and conserved regulatory mechanism for many metabolic enzymes in eukaryotes, prokaryotes, and even archaea (*Liu, 2016*; *Park and Horton, 2019*; *Zhou et al., 2020*). Here, we show two types of filament structures of *E. coli* PRPS. Between *E. coli* and human PRPS1/2, the hexameric propeller structure of PRPS and the identified residues responsible for polymerization are conserved, indicating that type A and / or type B PRPS filaments may also exist in mammals. In fact, PRPS isolated from rat liver tissues revealed a heterogenous protein complex with a molecular weight greater than 1000 kDa (*Kita et al., 1989*). A similar phenomenon was observed in human PRPS1/2 isolated from tissue sources (*Becker et al., 1977*; *Fox and Kelley, 1971*). Furthermore, the recombinant human PRPS1/2 purified from *E. coli* can also be spontaneously assembled into large complexes larger than 1000 kDa in vitro (*Nosal et al., 1993*). Although these complexes are not necessarily filaments depicted in this study, collective evidence suggests that class I PRPS is regulated by the assembly of large complexes in bacteria and mammals.

Filamentous polymers of some metabolic enzymes have been demonstrated to accommodate different states of proteins. In most cases, their protomers in various states are assembled through the same interface, which can enhance (e.g. human IMPDH, CTPS and *Drosophila* CTPS) or inhibit (e.g. *E. coli* CTPS) activity (*Johnson and Kollman, 2020*; *Lynch and Kollman, 2020*; *Zhou et al., 2021*; *Zhou et al., 2019b*). In other cases, enzymes can polymerize into multiple polymer types (e.g. human acetyl-CoA carboxylase) through different interfaces (*Hunkeler et al., 2018*). Here, we reveal two distinct PRPS filament types that show different regulatory functions (*Figure 8*). The protomer of type A filament is PRPS hexamer in closed conformation, while type B filament is assembled from hexamer in open conformation. Different contact interfaces are used to connect PRPS hexamers in each filament type (*Figures 5 and 6A*), this means that the hexamer conformation may be homogenous in the individual filament and also implies that the proportion of closed/open hexamers may determine the formation of either type of filaments over the other. PRPS hexamer conformation could be regulated by the binding of ligands. For instance, the binding of Pi at the allosteric site 1 and AMP/ADP

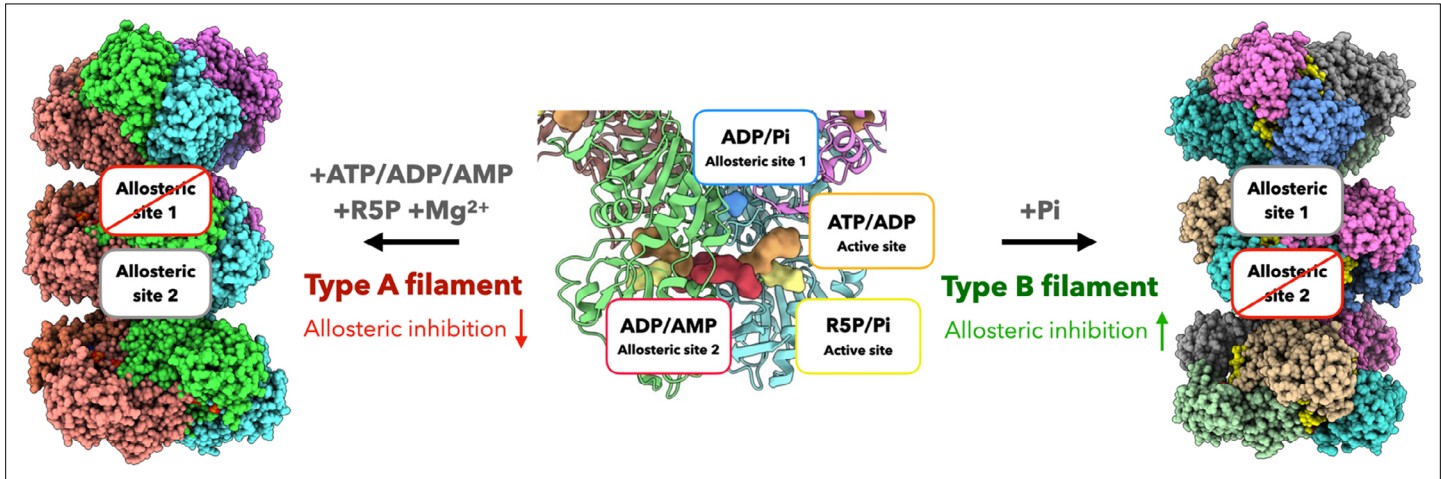

**Figure 8.** Schematic model of regulation and function of *E. coli* PRPS filaments. *E. coli* PRPS can form type A filament and type B filament to precisely regulate enzyme activity. ADP and phosphate (Pi) are allosteric regulators of PRPS. They can bind at allosteric sites or active sites. When incubated with ATP, ADP or AMP and R5P and Mg²⁺, PRPS can form type A filament to disrupt allosteric inhibition. When incubated with Pi, PRPS can form type B filament to increase allosteric inhibition.

at site 2 may promote the closed hexamer conformation, while the ADP at site 1 would prevent the open-to-closed transformation. In addition, it is reasonable to suspect that polymerization may also contribute to the conformation stability of hexamers. In a relatively simple environment, only one type of filament may be produced (*Figure 8*). We used a very high amount of Pi (50 mM) to induce type B filament. The substitution of R5P by Pi at the active site may not represent the physiological situation. However, our activity assay and live cell imaging suggest these two types of filaments coexist in more complex mixtures or cells. This mechanism provides a new PRPS regulatory layer that can fine tune the production of PRPP and coordinate cellular metabolic pathways.

ADP is an effective PRPS inhibitor because it can reduce the activity of rat PRPS by about 40~50% at a concentration of 0.1 mM (*Tatibana et al., 1995*). Consistent with previous studies (*Willemoës et al., 2000*), we showed that when 0.1 mM ADP was added into the reaction mixture, the PRPS^WT activity of *E. coli* decreased by about 40% (*Figure 6B and C*). It is not clear how ADP at allosteric site 1 reduces the activity, although a molecular model has been proposed by comparing the three-dimensional structures of various *B. subtilis* PRPP synthase complexes (*Hove-Jensen et al., 2017*). Unfortunately, our structural analysis could not provide additional information to this mechanism. However, we show that this allosteric inhibition by ADP could be compromised by the formation of type A filaments (*Figure 6B*). In addition, our data illustrate a clear model for the resistance mechanism. In this model, the RF loop moves toward allosteric site 1, thereby enclosing the ADP binding pocket. A similar mechanism has been depicted in a previous report on human PRPS1 crystal structures, in which the adjacent allosteric site 2 is bound by $SO_4^{2-}$ (*Li et al., 2007*). These suggest that PRPS filamentation is not essential for the regulatory function of allosteric site 2, but could enhance such an effect. Since intracellular ADP level is generally within the range of hundreds micromolar in *E. coli* and mammalian cells (*Meyrat and von Ballmoos, 2019*; *Traut, 1994*), it is reasonable to propose that type A filament facilitates PRPP production in cells (*Traut, 1994*). On the other hand, we find the RF loop in the type B filament is unstable. The new conformation of the RF loop in our model indicates that the conformational changes of this loop may also participate in the regulation of ATP binding and catalytic activity (*Figure 4B and C*). Collectively, our data demonstrate the important role of the RF loop in regulating PRPS activity in two ways.

In addition, our model also explains the regulation of conformational change of RF loop. In our type A filament model, additional allosteric site 2 binds to AMP/ADP (*Figure 3D* and *Figure 3—figure supplement 1*). AMP/ADP at this binding site interacts with the RF loop, thereby stabilizing the loop at allosteric site 1. In contrast, allosteric site 2 is incomplete in the PRPS open hexamer (type B filament) and cannot bind to AMP/ADP, resulting in an unstable RF loop (*Figure 4A and C*). The point mutation E133A is expected to impair allosteric site 2, resulting in a significant decrease in activity, which supports our hypothesis about the function of allosteric site 2.

PRPS mutants have been used to improve the production of commercial compounds with PRPP as intermediate. For instance, feedback-resistant mutant PRPS has been shown to increase the synthesis of riboflavin and purine nucleosides of *R. gossypii* and *B. amyloliquefaciens*, respectively (*Jiménez et al., 2008*; *Zakataeva et al., 2012*). Our findings reveal a new mechanism involved in the regulation of *E. coli* PRPS with structural basis. Considering the high conservation among class I PRPS, certain point mutations may be applicable to industry.

In humans, PRPS1 mutations lead to low or high activity of PRPP synthesis, which is related to various disorders. Some of these human PRPS1 mutants have been characterized at the molecular level (*Becker et al., 1995*). Inhibition kinetics indicated that they were not sensitive to the allosteric inhibition. Interestingly, mutations that lead to allosteric inhibition desensitization, such as D51H, L128I, D182H, A189V, and H192D, are almost exclusively located at the interface between the dimers in the hexamer (*Becker et al., 1995*). Whether these mutants are related to the abnormal conformations of open and closed hexamers require further investigation.

Cytoophidia are considered large filamentous polymer bundles of metabolic enzymes (*Liu, 2016*; *Park and Horton, 2019*). It has been suggested that many types of cytoophidia are correlated with specific cellular status in certain tissues. For instance, CTPS cytoophidia are widely distributed in *Drosophila* tissues, especially in proliferative cell types (*Aughey et al., 2014*; *Liu, 2010*; *Zhang et al., 2020b*). In mammals, CTPS cytoophidia have been found in mouse thymus and many human cancers (*Chang et al., 2017*; *Peng et al., 2021*). In addition to regulating enzyme activity, the cytoophidium may also protect component proteins from degradation (*Lin et al., 2018*; *Sun and Liu, 2019a*). Many

factors, such as mTOR pathway, temperature, pH, osmolality, and protein post-translational modifications, have been shown to influence the assembly of cytoophidia in various organisms (*Andreadis et al., 2019*; *Lin et al., 2018*; *Lynch and Kollman, 2020*; *Petrovska et al., 2014*; *Sun and Liu, 2019b*; *Wang et al., 2015*; *Zhang and Liu, 2019*). Recently, PRPS cytoophidia have been reported in yeast, *Drosophila*, zebrafish, and mammalian cell lines (*Begovich et al., 2020*; *Noree et al., 2019*). We demonstrate that PRPS cytoophidium is also present in *E. coli*, which may have physiological significance.

In conclusion, we show that *E. coli* PRPS can assemble two types of filaments. Structural comparison and biochemical analysis reveal a novel mechanism to regulate PRPS activity through conformational changes of RF loop, which is modulated by noncanonical allosteric site 2. These results expand our understanding of the regulation of key steps in nucleotide biosynthesis and shed light on potential clinical and industrial applications.

# Materials and methods

**Key resources table**

| Reagent type (species) or resource | Designation | Source or reference | Identifiers | Additional information |
|---|---|---|---|---|
| Gene (*Drosophila melanogaster*) | PRPS | Genbank | P0A717 | |
| Strain, strain background (*Escherichia coli*) | Transetta (DE3) | TransGen Biotech | | |
| Recombinant DNA reagent | pRSFDuet-6His | In house | | |
| Commercial assay or kit | BCA Protein Concentration Determination Kit (Enhanced) | Beyotime | P0010 | |
| Chemical compound, drug | Benzamidine hydrochloride | Sigma-Aldrich | 434760–5 G | |
| Chemical compound, drug | Pepstatin A | Sigma-Aldrich | P5318-25MG | |
| Chemical compound, drug | Leupeptin hydrochloride microbial | Sigma/Aldrich | L9783-100MG | |
| Chemical compound, drug | PMSF | MDBio | P006-5g | |
| Chemical compound, drug | Ni-NTA Agarose | QIAGEN | 30,250 | |
| Chemical compound, drug | Orotic acid | Adamas | 01102798 (74736A) | |
| Chemical compound, drug | ATP | Takara | 4,041 | |
| Chemical compound, drug | D-Ribose 5 phosphate disodium salt | BIOSYNTH CARBOSYNTH | R-5600 | |
| Chemical compound, drug | 5-phospho-D-ribose 1-diphosphate penta-sodium salt | Sigma | P8296-25 mg | |
| Chemical compound, drug | Adenosine 5'-monophosphate | solarbio | A9860-1 | |
| Chemical compound, drug | Adenosine 5'-diphosphate sodium salt | Sigma | A2754-100MG | |
| Other | Nitinol mesh | Zhenjiang Lehua Electronic Technology | M024-Au300-R12/13 | Cryo-EM grid preparation |
| Other | Holey Carbon Film | Quantifoil | R1.2/1.3, 300 Mesh, Cu | Cryo-EM grid preparation |
| Other | 400 mesh reinforced carbon support film | EMCN | BZ31024a | Negative staining |
| Software, algorithm | UCSF Chimera | https://doi.org/10.1002/jcc.20084 | | https://www.cgl.ucsf.edu/chimera |
| Software, algorithm | UCSF Chimera X | https://doi.org/10.1002/pro.3235 | | https://www.cgl.ucsf.edu/chimerax/ |
| Software, algorithm | Relion | https://doi.org/10.7554/eLife.42166 | | https://relion.readthedocs.io/en/latest/index.html# |
| Software, algorithm | Coot | https://doi.org/10.1107/S0907444910007493 | | https://www2.mrc-lmb.cam.ac.uk/personal/pemsley/coot/ |
| Software, algorithm | Phenix | https://doi.org/10.1107/S2059798318006551 | | https://phenix-online.org/ |

## Expression and purification of *E. coli* PRPS

Full-length of wild-type or mutant *E. coli* PRPS sequences with a C-terminal 6×His tag were cloned into a modified pRSFDuet vector at the MCS 2 site and expressed in *E. coli* Transetta (DE3) cells. After induction with 0.1 mM IPTG at the $OD_{600}$ range of 0.5~0.8, the cells were cultured at 37 °C for 4 hr and pelleted by centrifugation at 4000 r.p.m. for 20 min. All remaining purification procedures were performed at 4 °C. The harvested cells were resuspended in cold lysis buffer (50 mM Tris HCl pH8.0, 500 mM NaCl, 10% glycerol, 10 mM imidazole, 5 mM β-mercaptoethanol, 1 mM PMSF, 5 mM benzamidine, 2 µg/ml leupeptin and 2 µg/ml pepstatin). After ultrasonication, the cell lysate was then centrifuged (18,000 r.p.m.) at 4 °C for 45 min. The supernatant was collected and incubated with equilibrated Ni-NTA agarose beads (Qiagen) for 1 hr. Subsequently, the column was further washed with lysis buffer supplemented with 50 mM imidazole. Target proteins were eluted with elution buffer containing 50 mM Tris HCl pH8.0, 500 mM NaCl, 250 mM imidazole, and 5 mM β-mercaptoethanol. Further purification was performed in column buffer (25 mM Tris HCl pH 8.0 and 150 mM NaCl) using HiLoad Superdex 200 gel-filtration chromatography (GE Healthcare). The peak fractions were collected, concentrated, and stored in small aliquots at −80 °C.

## Cryo-EM grid preparation and data collection

To generate type A filaments, 6 µM PRPS protein was dissolved in a buffer containing 25 mM Tris HCl pH 7.5, 2 mM ATP, 10 mM $MgCl_2$. ATP was replaced with 2 mM ADP and 2 mM AMP for generating type $A^{AMP/ADP}$ filaments. For type B filaments formation, 6 µM PRPS protein was incubated in a buffer containing 50 mM $Na_2HPO_4$ and 100 mM NaCl. All samples were incubated at 37 °C for 30 minutes and then loaded onto the grid. In order to prepare low-temperature cryo-EM grids, protein samples were loaded on a 300-mesh amorphous alloy grids (CryoMatrix M024-Au300-R12/13) with fresh glow discharge. Grids were blotted for 3.5 s with blot force of −1 at 4 °C and 100% humidity before plunge-freezing in liquid ethane with an FEI Vitrobot Mark IV (ThermoFisher Scientific).

Micrographs were collected in super-resolution counting mode with K3 Summit direct electron camera (Gatan) on FEI Titan Krios electron microscope at 300 kV. Automated data acquisition was performed with SerialEM (*Mastronarde, 2005*) at a nominal magnification of 22,500×, corresponding to a calibrated pixel size of 1.06 Å with a defocus range from 1.0 to 2.5 µm. Each movie stack was acquired in a total dose of 60 $e^-$Å$^{-2}$, subdivided into 50 frames at 4 s exposure.

## image processing

All image processing steps were performed using Relion3.1-beta (*Zivanov et al., 2018*). Beam-induced motion correction and exposure weighting were performed by the MotionCorr2 (*Zheng et al., 2017*) and the CTF (contrast transfer function) parameter was estimated by CTFFIND4. For the type A filament dataset, 3045 images were manually selected and 887,654 particles were automatically picked up. Among them, after two rounds of fast 2D classification (extracting particles in binning 2) and another round of 2D classification (extracting particles in binning 1), 1438771 particles were selected for 3D classification. The featureless cylinder was reconstructed using the relion_helix_toolbox command and applied as a reference model for 3D classification. After two rounds of 3D classification using C1 and D3 symmetry, a total of 70,541 particles of the best category were selected for 3D auto-refinement, and each particle was subjected to CTF refinement and Bayesian polishing. Finally, the initial 2.8 Å density map including three layers of PRPS hexamer was ontained. A final 2.3 Å map was sharpened by post-process using a tight mask for the central hexamer with a B-factor of 45 Å$^2$.

A similar procedure was performed for the type B filament dataset. A total of 1,186,879 particles were auto-picked from 2776 images. After multiple rounds of 2D classification and 3D classification, 168,218 particles were selected for 3D auto-refinement, and each particle was subjected to CTF refinement and Bayesian polishing. By using a compact mask with a B-factor of 98 Å for the central hexamer$^2$, the optimal density map was sharpened to a nominal resolution of 2.9 Å.

For the type $A^{AMP/ADP}$ filament dataset, 1066797 particles were auto-picked from 1824 images. After multiple rounds of 2D classification and 3D classification, 53,045 particles were selected for 3D auto-refinement, and each particle is subjected to CTF refinement and Bayesian polishing. By using a compact mask with a B-factor of 51 Å for the central hexamer$^2$, the optimal density map was sharpened to a nominal resolution of 2.6 Å. LocalRes was used to estimate the local resolution for all maps.

## Model building and refinement

The Crystal structure of *E. coli* PRPS [Protein Data Bank (PDB) ID: 4S2U] was applied for the initial models of all datasets. The hexamer models were separated and docked into the corresponding electron density map using Chimera v.1.14 (*Pettersen et al., 2004*), followed by iterative manual adjustment and rebuilding in Coot (*Emsley and Cowtan, 2004*) and real-space refinement in PHENIX (*Adams et al., 2011*). The final atomic model was evaluated using MolProbity (*Williams et al., 2018*). The map reconstruction and model refinement statistics are listed in *Table 1*. All figures and videos were generated using UCSF Chimera and ChimeraX (*Goddard et al., 2018*).

## PRPS activity assay

On a 96-well plate, the activity of PRPS was measured by coupled continuous spectrophotometry using SpectraMax i3. The PRPS reaction (*ATP +R5 P → PRPP +AMP*) is coupled the forward reaction (OA +PRPP → OMP +PPi) of *E. coli* orotate phosphoribosyltransferase (OPRT, EC 2.4.2.10) and the amount of PRPP generated in the reaction was determined by the reduction of in orotate (OA) in the mixture. The concentration of OA was measured by absorbance at 295 nm for 300 s at 25 °C (*Krungkrai et al., 2005*). Reaction mixture (200 µl) contains 0.1 µM PRPS, 1 mM OPRT, 1 mM OA, 10 mM $MgCl_2$, 250 mM NaCl, 10 mM $Na_2HPO_4$, 0.6 mM R5P and AMP, ADP, ATP at concentrations as described in each experiment. ATP or R5P was least added into the mixture to initiate the reaction. All measurements were performed in triplicate.

## Negative staining electron microscopy

The purified *E. coli* PRPS protein (1 µM) was dissolved in Tris-HCl buffer (25 mM Tris-HCl, 150 mM NaCl, 10 mM $MgCl_2$, and 2 mM ATP or ADP) or Pi buffer (50 mM $Na_2HPO_4$, 300 mM NaCl). After incubation at 37 °C for 1 hr, the protein samples were loaded onto hydrophilic carbon-coated grids and washed twice with uranium formate. Subsequently, the grids were stained with uranium formate. Imaging was acquired with 120 kV electron microscope (Talos L120C, ThermoFisher, USA) with Eagle 4 K × 4 K CCD camera system (Ceta CMOS, ThermoFisher, USA) at ×57,000 magnification.

## Sample preparation and confocal microscopy

*E. coli* cells were fixed with 4% formaldehyde at 37 °C, 220 r.p.m. for 10 min. After fixation, cells were collected by centrifugation at 12,000 r.p.m. for 1 min. Cell pellets were washed twice with PBS and then resuspended in PBS containing Hoechst33342. The cells were then incubated at room temperature for 1 hr. Add 2.5 µL of cell solution and 1 µL of pre-melted 1.2% low melting-point agar was mixed on the glass slide and covered with a coverslip for observation. Images were captured under Plan-Apochromat 63×/1.40 Oil DIC M27 objective on a Carl Zeiss LSM 800 (Axio Observer Z1) inverted fluorescence confocal microscope.

## Growth curve

*E. coli* cells were pre-cultured overnight in 2 mL LB medium at 37 °C, 220 r.p.m., and then inoculated in 5 mL LB culture at 37 °C, 220 r.p.m. with $OD_{600}$=0.05. The cell growth was determined by the OD600 value, which was measured by Eppendorf BioPhotometer D30 every hour after inoculation.

## Acknowledgements

EM data were collected at the ShanghaiTech Cryo-EM Imaging Facility. We thank the Molecular and Cell Biology Core Facility (MCBCF) at the School of Life Science and Technology, ShanghaiTech University for providing technical support. This work was supported by grants from Ministry of Science and Technology of China (No. 2021YFA0804701-4), National Natural Science Foundation of China (No. 31771490), Shanghai Science and Technology Commission (No. 20JC1410500) and the UK Medical Research Council (grant nos. MC_UU_12021/3 and MC_U137788471) for grants to J.L.L.

## Additional information

### Funding

| Funder | Grant reference number | Author |
|---|---|---|
| Ministry of Science and Technology of the People's Republic of China | 2021YFA0804701-4 | Ji-Long Liu |
| National Natural Science Foundation of China | 31771490 | Ji-Long Liu |
| Shanghai Science and Technology Commission | 20JC1410500 | Ji-Long Liu |
| Medical Research Council | MC_UU_12021/3 | Ji-Long Liu |
| Medical Research Council | MC_U137788471 | Ji-Long Liu |

The funders had no role in study design, data collection and interpretation, or the decision to submit the work for publication.

### Author contributions

Huan-Huan Hu, Conceptualization, Data curation, Formal analysis, Investigation, Visualization, Writing – original draft, Writing – review and editing; Guang-Ming Lu, Conceptualization, Formal analysis, Investigation, Validation, Writing – original draft; Chia-Chun Chang, Conceptualization, Investigation, Writing – original draft, Writing – review and editing; Yilan Li, Jiale Zhong, Chen-Jun Guo, Xian Zhou, Boqi Yin, Tianyi Zhang, Investigation; Ji-Long Liu, Conceptualization, Funding acquisition, Project administration, Supervision, Writing – original draft, Writing – review and editing

### Author ORCIDs

Huan-Huan Hu ⓘ http://orcid.org/0000-0002-5997-530X
Guang-Ming Lu ⓘ http://orcid.org/0000-0002-6607-2264
Chia-Chun Chang ⓘ http://orcid.org/0000-0001-7942-6300
Yilan Li ⓘ http://orcid.org/0000-0002-4355-6522
Jiale Zhong ⓘ http://orcid.org/0000-0001-5873-0450
Chen-Jun Guo ⓘ http://orcid.org/0000-0001-5342-4761
Xian Zhou ⓘ http://orcid.org/0000-0002-0000-2415
Boqi Yin ⓘ http://orcid.org/0000-0003-3974-9820
Tianyi Zhang ⓘ http://orcid.org/0000-0002-4632-6298
Ji-Long Liu ⓘ http://orcid.org/0000-0002-4834-8554

### Decision letter and Author response

Decision letter https://doi.org/10.7554/eLife.79552.sa1
Author response https://doi.org/10.7554/eLife.79552.sa2

---

## Additional files

### Supplementary files

• MDAR checklist

### Data availability

Atomic models generated in this study have been deposited at the PDB under the accession codes 7XMU, 7XMV, 7XN3. Cryo-EM maps deposited to EMDB as: EMD-33305, EMD-33306, EMD-33309.

The following datasets were generated:

| Author(s) | Year | Dataset title | Dataset URL | Database and Identifier |
|-----------|------|---------------|-------------|-------------------------|
| Hu HH, Lu GM, Chang CC, Liu JL | 2022 | *E.coli* phosphoribosylpyrophosphate (PRPP) synthetase type A filament bound with ADP, Pi and R5P | https://www.rcsb.org/structure/7XMU | RCSB Protein Data Bank, 7XMU |
| Hu HH, Lu GM, Chang CC, Liu JL | 2022 | *E.coli* phosphoribosylpyrophosphate (PRPP) synthetase type B filament bound with Pi | https://www.rcsb.org/structure/7XN3 | RCSB Protein Data Bank, 7XN3 |
| Hu HH, Lu GM, Chang CC, Liu JL | 2022 | *E.coli* phosphoribosylpyrophosphate (PRPP) synthetase type A(AMP/ADP) filament bound with ADP, AMP and R5P | https://www.rcsb.org/structure/7XMV | RCSB Protein Data Bank, 7XMV |
| Hu HH, Lu GM, Chang CC, Liu JL | 2022 | *E.coli* phosphoribosylpyrophosphate (PRPP) synthetase type A filament bound with ADP, Pi and R5P | https://www.ebi.ac.uk/emdb/EMD-33305 | Electron Microscopy Data Bank, EMD-33305 |
| Hu HH, Lu GM, Chang CC, Liu JL | 2022 | *E.coli* phosphoribosylpyrophosphate (PRPP) synthetase type A (AMP/ADP) filament bound with ADP, AMP and R5P | https://www.ebi.ac.uk/emdb/EMD-33306 | Electron Microscopy Data Bank, EMD-33306 |
| Hu HH, Lu GM, Chang CC, Liu JL | 2022 | *E.coli* phosphoribosylpyrophosphate (PRPP) synthetase type B filament bound with Pi | https://www.ebi.ac.uk/emdb/EMD-33309 | Electron Microscopy Data Bank, EMD-33309 |

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
