## [Editor Report]

This paper provides new insights into how polymerization into two different structures modulates the activity of the enzyme PRPS. The molecular mechanisms proposed are supported by the data, and likely to be of general interest.

---

## [Decision Letter]

**Decision letter after peer review:**

Thank you for submitting your article "Allosteric inhibition of PRPS is moderated by filamentous polymerization" for consideration by *eLife*. Your article has been reviewed by 3 peer reviewers, one of whom is a member of our Board of Reviewing Editors, and the evaluation has been overseen by Philip Cole as the Senior Editor. The reviewers have opted to remain anonymous.

Essential revisions:

While all three reviewers found the work significant and of potential broad interest, all felt that the paper needed major rewriting to improve the clarity and language. Numerous examples are given below where improvements are needed.

*Reviewer #1 (Recommendations for the Authors):*

The paper needs careful rewriting, and authors should check and review extensively for improvements to the use of English. Here are just three examples out of many that I found:

(line 57) "…outcomes in clinical."

(line 73) "the ADP" should be "ADP".

(line 352-4) "We fused the sequences of PRPSWT and PRPSR302A, PRPSY24A and PRPSR302A/Y24A mutants were with mCherry and overexpressed them in *E. coli* individually."

*Reviewer #2 (Recommendations for the Authors):*

The manuscript reports impressive high-quality cryo-EM reconstructions of two different PRPP synthetase filaments. The detailed structural analysis, combined with structure-based mutagenesis and enzymatic assays reveals a quite complex regulatory mechanism for this important enzyme. I realize the difficulty of the study, given that ADP can bind to three different sites with different effects on the protein activity, and this makes the analysis very difficult. The finding that the assembly of the enzyme in two different filamentous structures and the repositioning of the regulatory loop can affect the binding of ADP to the different regulatory sites, and the binding of the substrates is very interesting.

I encourage the authors to review carefully the manuscript to facilitate the comprehension of the main message.

*Reviewer #3 (Recommendations for the Authors):*

1. Some language editing would aid in the clarity of the text (throughout the manuscript).

2. Species names should be italicized.

3. Page 5, line 85: "The cytoophidium is assembled by bunding filamentous polymers of metabolic enzymes" – is this known for all cytoophidia? If so, a reference should be given. If not, the sentence can be rewritten to indicate that this is speculation or based on some evidence in some cases.

4. Page 6, line 94: the sentence "The polymerizations of CTPS and IMPDH have been demonstrated to desensitize the proteins to end-product inhibition or allosteric inhibition effects, suggesting its physiological functions in tuning intracellular nucleotide levels". Please provide a reference to support this statement. Also, is this true in all cases of CTPS/IMPDH polymerization?

5. Page 9, line 150: what was the concentration of Mg^2+^ used in the structure with 2 mM ATP (and no π I presume?)

6. Page 9, lines 157 and 167, please state the handedness of the twist of the polymers.

7. Page 9, line 157, it is stated that R5P is found in the active site, though it was not added to the enzyme before structural studies were conducted. The authors suggest that it may have copurified with the PRPS, but it is not present in the Type B filament structure. The authors also do not show the map around the R5P. The map should be shown to provide confidence in the assignment of R5P to the map in the active site. Could it be formed from the degradation of ATP (just as the ADP and π presumably are?).

8. page 9, line 163 and elsewhere: it's awkward to refer to the structure of L. pneumophila as a model. It should be referred to as an x-ray crystal structure (if indeed that is what it is).

9. Figure 2 and text regarding panels in figure 2. It is difficult to deconvolute all the structures and alignments without referencing the figure legend constantly. It would be easier if structures were identified in the panels, such as by color coding their name/type in the panel where they are shown in particular colors. When colors are ambiguous (such as in panel 2e, a label and arrow would aid in distinguishing the type A from the type B conformation of the RF loop). In the text, when a particular structure is being indicated, the color of that structure in addition to the panel id should be given (e.g. "yellow, Figure 2c").

10. Page 10, line 169: "ATP binding site and the allosteric site 1 are not bound by any ligand". Should this read "allosteric site 2"?

11. Page 10, line 183: some object to the use of the word "monomer" in the context of an oligomer. A protein chain or subunit or protomer would be more appropriate.

12. It is interesting that the allosteric sites and active sites are found between protein chains in the hexamer, yet this isn't mentioned.

13. page 11, line 190. "His131 and the phosphate". please indicate which phosphate (it appears to be the β).

14. page 11, 205-206. regarding the second type A filament structure, which was incubated apparently in both AMP and ADP, how is it determined that both AMP and ADP are bound in site 2? Was refinement done with partial occupancy of each? The sentence describing this structure indicates at first that AMP is in site 2, then the next sentence indicates that ADP is bound. Which is it? this should be more clear.

15. page 11, line 208. should guanine be adenine?

16. page 11, line 211, "and residues responsible for the binding", please indicate which residues. Also, in figure 2, highlight the residues suggested to be conserved. Residues suggested to be involved in binding to ligands include 102, 178, 182, 149, 133, and 147, however, 178, 182, and 149 do not appear to be conserved. Also, are residues of site 1 conserved? if not, what would that mean?

17. page 12, lines 229 and 231, why are the B factors negative?

18. page 12, line 232, should "figure 2f" be figure 2e?

19. Page 12, line 218, it should be indicated that allosteric site 1 is the site being discussed.

20. page 12, line 220. Although ATP binds only in the active site, it should be indicated whenever the ATP binding site is mentioned that it is the ATP binding site in the active site (since it is hard to keep track of which site binds which type of nucleotide).

21. page 12, line 220, which phosphate binding site of R5P does π bind at?

22. page 12, line 233, "In the stable conformation", is this of the type A filament? If so, it should be indicated.

23. Use of apostrophes or asterisks on the residue names in the second paragraph (beginning at line 244) of page 13 would help to show that residues are contributed by different subunits.

24. page 13, line 250, indicate which atom of Leu23 hydrogen bonds to Arg301 (presumably the carbonyl oxygen of the main chain of Leu23?).

25. page 15, line 280: indicate panel c of Sup 5. Also, what are the concentrations of ligands kept at constant concentrations in Sup 5? (e.g. the concentration of ATP when R5P is varied?). What is the concentration of the enzyme?

26. page 16, line 315, "the allosteric inhibition", indicate that it is by ADP, i.e. "allosteric inhibition by ADP".

27. page 16, line 319, "enhance the inhibitory effect" to "enhance the inhibitory effect of ADP".

28. page 17, line 328, "Since AMP is supposedly unable to bind at the allosteric site 1", please provide a reference and/or rationale to support this assertion.

29. page 21, line 400, "Although some filamentous polymers of metabolic enzymes were demonstrated to accommodate different states of the proteins, their protomers in various states assemble through the same interfaces and can either boost or suppress the activity". References for CTPS are given here although the interfaces in CTPS filaments that are activated differ from those which are inhibited. This is true when CTPS from different organisms are compared. Please correct or clarify what is meant by this sentence as it appears to indicate that for CTPS, one type of filament forms which can accommodate either active or inactive conformations. Further, there are other enzymes for which different interfaces result in different filamentous forms, with differing activity, for example, ACC enzyme.

30. page 22, line 432: "According to our models, both allosteric site 1 and site 2 could be responsible for the binding of the second ADP in these conditions". It seems as though the conclusion of the paper is that ADP binding at allosteric site 1 would block ATP binding to the active site. Is this not true?

31. Page 38, line 689, please indicate allosteric site 1 in "while the allosteric site".

32. Colors in Figure 2, panels d and f need to be defined in the figure legend.

33. page 40, line 708: "negative staining" should read "negative staining electron microscopy" (and elsewhere in the manuscript).

34. Please define the meaning of ***, and so on in the figures.

35. Please define "S.E.M." in figure 3e legend.

36. figure 4b, why is "n.s." shown for the orange line (the double mutant with mCherry)? It is true that the growth of this mutant is similar to wild type without mCherry, but it is much less when compared to the growth of wild type with mCherry.

37. π is found in the type A filament, yet it is absent from the type A filament shown in panel 4c. It also competes for binding with ADP in site 1. Should the "+Pi" be located with the other arrow, to type A, and not type B?

38. In Figure 4a, the text of the superscripts is very small.

39. Please provide some explanation of the model shown in figure 4c in the figure legend.

40. Please provide the concentrations of ligands (AMP, ADP, ATP, etc) in each image of Figure S1.

41. Please indicate somehow that the phosphate ion shown in Figure S2c is derived only from the type A filament (and not the type A^AMP/ADP^).

---

## [Author Response]

Reviewer #1 (Recommendations for the Authors):The paper needs careful rewriting, and authors should check and review extensively for improvements to the use of English. Here are just three examples out of many that I found:(line 57) "…outcomes in clinical."(line 73) "the ADP" should be "ADP".(line 352-4) "We fused the sequences of PRPSWT and PRPSR302A, PRPSY24A and PRPSR302A/Y24A mutants were with mCherry and overexpressed them in *E. coli* individually."

We have rewritten the manuscript to improve the clarity and presentation.

Reviewer #3 (Recommendations for the Authors):1. Some language editing would aid in the clarity of the text (throughout the manuscript).

We have rewritten the manuscript to improve the clarity and presentation.

2. Species names should be italicized.

Revised.

3. Page 5, line 85: "The cytoophidium is assembled by bunding filamentous polymers of metabolic enzymes" – is this known for all cytoophidia? If so, a reference should be given. If not, the sentence can be rewritten to indicate that this is speculation or based on some evidence in some cases.

By definition, all cytoophidia are bundled filamentous polymers of metabolic enzymes. Related references have been added.

4. Page 6, line 94: the sentence "The polymerizations of CTPS and IMPDH have been demonstrated to desensitize the proteins to end-product inhibition or allosteric inhibition effects, suggesting its physiological functions in tuning intracellular nucleotide levels". Please provide a reference to support this statement. Also, is this true in all cases of CTPS/IMPDH polymerization?

Indeed, some differences may present among species or protein isoforms. We specified human CTPS1 and IMPDH2 in this sentence and added references.

5. Page 9, line 150: what was the concentration of Mg^2+^ used in the structure with 2 mM ATP (and no π I presume?)

The concentration of Mg^2+^ has been added in the text. No π in this condition.

6. Page 9, lines 157 and 167, please state the handedness of the twist of the polymers.

Added (left-handed twist) in the description.

7. Page 9, line 157, it is stated that R5P is found in the active site, though it was not added to the enzyme before structural studies were conducted. The authors suggest that it may have copurified with the PRPS, but it is not present in the Type B filament structure. The authors also do not show the map around the R5P. The map should be shown to provide confidence in the assignment of R5P to the map in the active site. Could it be formed from the degradation of ATP (just as the ADP and π presumably are?).

We have added a supplementary figure showing the map around R5P.

8. page 9, line 163 and elsewhere: it's awkward to refer to the structure of L. pneumophila as a model. It should be referred to as an x-ray crystal structure (if indeed that is what it is).

Revised.

9. Figure 2 and text regarding panels in figure 2. It is difficult to deconvolute all the structures and alignments without referencing the figure legend constantly. It would be easier if structures were identified in the panels, such as by color coding their name/type in the panel where they are shown in particular colors. When colors are ambiguous (such as in panel 2e, a label and arrow would aid in distinguishing the type A from the type B conformation of the RF loop). In the text, when a particular structure is being indicated, the color of that structure in addition to the panel id should be given (e.g. "yellow, Figure 2c").

Revised.

10. Page 10, line 169: "ATP binding site and the allosteric site 1 are not bound by any ligand". Should this read "allosteric site 2"?

Corrected.

11. Page 10, line 183: some object to the use of the word "monomer" in the context of an oligomer. A protein chain or subunit or protomer would be more appropriate.

Revised.

12. It is interesting that the allosteric sites and active sites are found between protein chains in the hexamer, yet this isn't mentioned.

We have now mentioned in the text.

13. page 11, line 190. "His131 and the phosphate". please indicate which phosphate (it appears to be the β).

It is the α phosphate. Revised accordingly.

14. page 11, 205-206. regarding the second type A filament structure, which was incubated apparently in both AMP and ADP, how is it determined that both AMP and ADP are bound in site 2? Was refinement done with partial occupancy of each? The sentence describing this structure indicates at first that AMP is in site 2, then the next sentence indicates that ADP is bound. Which is it? this should be more clear.

In type A^AMP/ADP^ filament, only AMP is bound to site 2. We revise the sentence to avoid confusion.

15. page 11, line 208. should guanine be adenine?

Corrected.

16. page 11, line 211, "and residues responsible for the binding", please indicate which residues. Also, in figure 2, highlight the residues suggested to be conserved. Residues suggested to be involved in binding to ligands include 102, 178, 182, 149, 133, and 147, however, 178, 182, and 149 do not appear to be conserved. Also, are residues of site 1 conserved? if not, what would that mean?

We revised the figure accordingly (New Figure 3 and Figure 3 —figure supplement 2).

17. page 12, lines 229 and 231, why are the B factors negative?

It should be positive. Corrected.

18. page 12, line 232, should "figure 2f" be figure 2e?

Corrected.

19. Page 12, line 218, it should be indicated that allosteric site 1 is the site being discussed.

Modified.

20. page 12, line 220. Although ATP binds only in the active site, it should be indicated whenever the ATP binding site is mentioned that it is the ATP binding site in the active site (since it is hard to keep track of which site binds which type of nucleotide).

Modified.

21. page 12, line 220, which phosphate binding site of R5P does π bind at?

Pi binds at the 5-phosphate binding site of R5P.

22. page 12, line 233, "In the stable conformation", is this of the type A filament? If so, it should be indicated.

The sentence is modified.

23. Use of apostrophes or asterisks on the residue names in the second paragraph (beginning at line 244) of page 13 would help to show that residues are contributed by different subunits.

Modified.

24. page 13, line 250, indicate which atom of Leu23 hydrogen bonds to Arg301 (presumably the carbonyl oxygen of the main chain of Leu23?).

Added.

25. page 15, line 280: indicate panel c of Sup 5. Also, what are the concentrations of ligands kept at constant concentrations in Sup 5? (e.g. the concentration of ATP when R5P is varied?). What is the concentration of the enzyme?

We added detail description of the reaction mixtures in the text and legends.

26. page 16, line 315, "the allosteric inhibition", indicate that it is by ADP, i.e. "allosteric inhibition by ADP".

The paragraph has been modified.

27. page 16, line 319, "enhance the inhibitory effect" to "enhance the inhibitory effect of ADP".

The paragraph has been modified.

28. page 17, line 328, "Since AMP is supposedly unable to bind at the allosteric site 1", please provide a reference and/or rationale to support this assertion.

This sentence is removed to avoid confusion.

29. page 21, line 400, "Although some filamentous polymers of metabolic enzymes were demonstrated to accommodate different states of the proteins, their protomers in various states assemble through the same interfaces and can either boost or suppress the activity". References for CTPS are given here although the interfaces in CTPS filaments that are activated differ from those which are inhibited. This is true when CTPS from different organisms are compared. Please correct or clarify what is meant by this sentence as it appears to indicate that for CTPS, one type of filament forms which can accommodate either active or inactive conformations. Further, there are other enzymes for which different interfaces result in different filamentous forms, with differing activity, for example, ACC enzyme.

We revised the sentences to

“Filamentous polymers of some metabolic enzymes have been demonstrated to accommodate different states of proteins. In most cases, their protomers in various states are assembled through the same interface, which can enhance (e.g. human IMPDH, CTPS and *Drosophila* CTPS) or inhibit (e.g. *E. coli* CTPS) activity (Johnson and Kollman, 2020; Lynch and Kollman, 2020; Zhou et al., 2021; Zhou et al., 2019b). In other cases, enzymes can polymerize into multiple polymer types (e.g. human acetyl-CoA carboxylase) through different interfaces (Hunkeler et al., 2018)”.

30. page 22, line 432: "According to our models, both allosteric site 1 and site 2 could be responsible for the binding of the second ADP in these conditions". It seems as though the conclusion of the paper is that ADP binding at allosteric site 1 would block ATP binding to the active site. Is this not true?

We removed the confusing sentence.

31. Page 38, line 689, please indicate allosteric site 1 in "while the allosteric site".

Added.

32. Colors in Figure 2, panels d and f need to be defined in the figure legend.

Corrected.

33. page 40, line 708: "negative staining" should read "negative staining electron microscopy" (and elsewhere in the manuscript).

Corrected.

34. Please define the meaning of ***, and so on in the figures.

Added **p* < 0.05, ** *p* < 0.01 and *** *p* < 0.001. in figure legends.

35. Please define "S.E.M." in figure 3e legend.

Edited to Error bars = standard error of the mean (S.E.M.).

36. figure 4b, why is "n.s." shown for the orange line (the double mutant with mCherry)? It is true that the growth of this mutant is similar to wild type without mCherry, but it is much less when compared to the growth of wild type with mCherry.

Here “wild type with mCherry” means “overexpression of wild-type PRPS fused with mCherry”. We revised figure legends accordingly.

37. π is found in the type A filament, yet it is absent from the type A filament shown in panel 4c. It also competes for binding with ADP in site 1. Should the "+Pi" be located with the other arrow, to type A, and not type B?

We added “+Pi” to the arrow to type A.

38. In Figure 4a, the text of the superscripts is very small.

Modified.

39. Please provide some explanation of the model shown in figure 4c in the figure legend.

Added.

40. Please provide the concentrations of ligands (AMP, ADP, ATP, etc) in each image of Figure S1.

Added.

41. Please indicate somehow that the phosphate ion shown in Figure S2c is derived only from the type A filament (and not the type A^AMP/ADP^).

We added the following sentence to explain why there is no π in the type A^AMP/ADP^ model:

“In addition, the allosteric site 1 in type A^AMP/ADP^ model is empty, supporting that the presence of ADP and π in the former type A filament model were due to spontaneous hydrolysis of ATP”.